# The mediating role of self-efficacy in the relationship between self-management and health-promoting behaviors in post-PCI patients

Zhijie Cao[1], Ping Yan[1], Fang Hou[2], Xin Gu[1], Hu Peng[1], Lina Ma[1], Li Zhang[3,4]*

1 School of Nursing, Xinjiang Medical University, Urumqi, Xinjiang, China, 2 The First Affiliated Hospital of Xinjiang Medical University, Urumqi, Xinjiang, China, 3 Department of Nursing, The First Affiliated Hospital of Xinjiang Medical University, State Key Laboratory of Pathogenesis, Prevention and Treatment of High Incidence Diseases in Central Asia, Urumqi, China, 4 Health Care Research Center for Xinjiang Regional population, Urumqi, Xinjiang, China

* 15684885009@163.com (LZ); Cao21119@163.com (ZC)

## Abstract

### Background

Coronary artery disease (CAD) has become one of the most prevalent diseases worldwide. Self-management and self-efficacy are critical for enhancing healthy lifestyles in patients with postoperative CAD and are strongly associated with health-promoting behaviors. However, the underlying mechanisms of this association remain unclear.

### Purpose

To explore the mediating role of self-efficacy between self-management and health-promoting behaviors in patients following percutaneous coronary intervention (PCI).

### Methods

From November 2024 to March 2025, 400 post-PCI patients were selected by convenience sampling from a tertiary first-class hospital in Xinjiang. Surveys were administered using a general information questionnaire, a self-management scale, a self-efficacy scale, and a health-promoting lifestyle scale. Structural equation modelling was employed to investigate the pathways of action among the three variables.

### Results

The scores for self-management behaviour, self-efficacy, and health-promoting lifestyle were (96.66 ± 10.11) scores, (44.49 ± 6.10) scores, and (162.91 ± 12.24) scores, respectively, with positive correlations between each pair of items. Self-efficacy partially mediated the relationship between self-management and health-promoting

**Data availability statement:** The minimal dataset underlying the results cannot be shared publicly due to ethical restrictions protecting patient confidentiality. De-identified data are available upon reasonable request to qualified researchers. Requests should be directed to the Institutional Review Board of the First Affiliated Hospital of Xinjiang Medical University ( Address:No. 137 Liyushan South Road, Urumqi, Xinjiang Uygur Autonomous Region, P.R. China. Email: dhl0030@163.com). The data are held securely by the institution for long-term availability per its data retention policies.

**Funding:** This work were supported by the Project of Cultivation of Excellence Talents and Innovative Teams of the First Affiliated Hospital of Xinjiang Medical University (cxtd202414) Xinjiang Key Laboratory of Medical Animal Model Research, and the State Key Laboratory of Pathogenesis, Prevention and Treatment of High Incidence Diseases in Central Asia Fund(SKL-HIDCA-2023-HL4. The funders had no role in study design, data collection and analysis, decision to publish, or preparation of the manuscript.

**Competing interests:** The authors have declared that no competing interests exist.

behaviors, with a standardized indirect effect of 0.317, accounting for 61.3% of the total effect.

## Conclusions

Self-efficacy plays a partial mediating role between self-management and health-promoting behaviors. Healthcare professionals should pay attention to the self-efficacy of patients post-PCI to enhance their awareness of personal health and strengthen self-management, thereby promoting proactive adoption of healthy lifestyles.

## Introduction

Coronary artery disease (CAD) is a leading cause of disability and mortality, imposing a substantial burden on healthcare systems globally [1–2]. Percutaneous coronary intervention (PCI) is a method used to enhance myocardial blood circulation by unblocking narrowed or occluded coronary artery lumens with a cardiac catheter. PCI has the advantages of minimal invasiveness, short operation time, and quick postoperative recovery, making it the primary clinical diagnostic and treatment tool for patients with CAD [3]. However, PCI cannot slow down or reverse the progression of atherosclerosis, nor can it eliminate cardiovascular safety risks [4–5]. Up to 2–10% of patients may experience in-stent restenosis after PCI [6]. Furthermore, post-PCI patients often have complex CAD lesions, making them susceptible to disease recurrence. A lack of relevant knowledge can lead to increased psychological burden and reduced quality of life [7–8].Therefore, it is critical to enhance the self-management capacity and lifestyle of patients with CAD.

Self-management is the ability and process through which individuals manage their behaviour and control disease through cognitive and behavioural strategies [9]. Several studies have shown that self-management is essential for reducing risk factors, improving symptoms, and enhancing the quality of life for patients with CAD [10–12].Self-management in chronic diseases may help patients manage symptoms, improve quality of life, and reduce their healthcare burden [13].It has a direct impact on prognosis, re-hospitalisation rates, and the somatic functioning of patients [14–15], and self-efficacy is the main determinants of self-management [16]. Health – promoting behaviors cover a broader range of lifestyle optimizations, such as nutrition management, stress regulation, social support, and life appreciation [17]. Although there is an overlap in behaviors between the two, their dimensions of action are different: the former focuses on disease – specific coping, while the latter emphasizes the maintenance of overall health. This distinction is particularly important for the management of patients after PCI. Patients need to accurately implement disease management and also establish an overall healthy ecosystem.Individual-level self-management involves the adoption of healthy behaviors and proactive measures to prevent disease progression [9],managing risk factors through lifestyle changes is essential for enhancing survival rates and reducing the recurrence of cardiovascular events [11].

Self-efficacy is defined as the information and personal initiative involved in exercising control. [18] For patients with CAD, self-efficacy can influence their psychological status and quality of life through positive emotions [19]. According to Willis et al. [20], self-efficacy is a major factor influencing individuals' attitudes, confidence, and behaviors towards the disease and is important for enhancing the quality of life of patients with CAD. Although self-efficacy has been widely studied in chronic disease management, patients after PCI face a "knowledge-behavior" translation dilemma. Due to the alleviation of disease symptoms after PCI, patients often underestimate the long-term risk of cardiovascular events, leading to a gradual decline in the use rate of cardiovascular preventive medications over time [21]. Most existing studies focus on single behavior changes and have not clarified how self-efficacy mediates the complex interaction between self-management and health-promoting behaviors during the dynamic rehabilitation process.

Therefore, there may be an association between self-management, self-efficacy, and health-promoting behaviors; however, the relationships among these factors and the pathways of action have not yet been fully determined. This study aimed to investigate self-management, self-efficacy, and health-promoting behaviors among postoperative patients who have received PCI, and establish a structural equation model to identify the mediating variables and interaction effects through which self-management influence health-promoting behaviors. The ultimate goal is to provide a theoretical basis for enhancing health-promoting behaviors within this population.

## Methods

### Participants and samples

Patients who underwent PCI at a grade 3A hospital in Xinjiang from November 2024 to March 2025 were recruited through convenience sampling. A total of 400 patients post-PCI were selected as study participants.

Inclusion criteria: (1) patients who met the diagnostic criteria outlined in the Chinese Guidelines for Percutaneous Coronary Intervention (2016) and underwent successful PCI [22]; (2) patients aged ≥18 years; and (3) patients who provided informed consent to voluntarily participate in this study.

Exclusion criteria: (1) patients with cognitive or mental disorders or visual and hearing impairments; (2) patients with functional impairment of vital organs or serious chronic diseases affecting other systems; (3) patients with language dysfunction that hindered normal communication.

The total number of independent variables in this study was 27, with the sample size calculated as 10–20 times the number of independent variables in the questionnaire. The initial sample size was 297–594. Considering a loss-to-follow-up rate of 10%, the final sample size was determined to be 400.

### Survey tools

#### Demographics and characteristics

Independently designed by the investigators, the measurement includes: (1) general demographic information, such as the patients' sex, age, educational level, marital status, monthly household income, occupational status, smoking status, and alcohol consumption; (2) Disease-related data, including whether the patient is overweight, whether it is the first diagnosis of CAD, the presence of comorbidities, and cardiac function grading.

#### Self-management behaviour scale

The self-management behaviour of patients with postoperative CAD was assessed using a modified version of the self-management behaviour scale developed by Hu [23], based on the original scale created by Ren [24]. This modification followed a comprehensive review of the existing scale. The scale comprised a total of 27 items organised into 7 dimensions: management of unhealthy habits, management of daily life, management of symptoms, management of disease knowledge, management of treatment adherence, management of first aid, and management of emotions and

cognition. A 5-point Likert scale was employed (1 = never, 2 = almost never, 3 = sometimes, 4 = often, 5 = always), the scale scores range from 0 to 135, with scores below 81 indicating a low level of self-management behavior, scores between 81 and 107 indicating a moderate level, and scores between 108 and 135 indicating a high level of self-management behavior,with higher scores indicating better self-management behaviors.The Cronbach's alpha coefficient for the scale was 0.913, while the Cronbach'salpha coefficient for the scale in this study was 0.735.The scale has demonstrated good content validity and construct validity in studies involving Chinese CAD patients [25–26].

### Self-efficacy scale

The Chinese version of the Cardiac Self-Efficacy Scale, developed by Sullivan et al. [27] and translated by Xie et al [28]. was utilised to assess the self-efficacy of patients with CAD. The scale is divided into two dimensions: maintaining function and controlling symptoms, comprising a total of 16 items. A 5-point Likert scale was employed (0 = no confidence at all, 1 = partial confidence, 2 = moderate confidence, 3 = very confident, 4 = highly confident), scores below 39 indicate a low level of self-efficacy, scores between 39 and 51 indicate a moderate level of self-efficacy, and scores above 51 indicate a high level of self-efficacy,with higher scores indicating greater self-efficacy among patients. The Cronbach's alpha coefficient for the scale was 0.820 while the Cronbach's alpha coefficient for the scale in this study was 0.640.The scale has demonstrated good content validity and construct validity in studies involving Chinese CAD patients [25–26].

### Health-promoting lifestyle scale

The Health-Promoting Lifestyle Scale-II, developed by Walker et al [29], was utilised to assess the level of health-promoting behaviors among the study participants. The scale comprised six dimensions and 52 items, including stress management (8 items), exercise (8 items), interpersonal support (9 items), nutrition (9 items), health responsibility (9 items), and self-actualisation (9 items). The scale was scored using a graded system with a score range of 52–208. Scores of 52–90 indicate a poor level, 91–129 a general level, 130–168 a good level, and 169–208 an excellent level.A higher score indicated better health-promoting behaviors. The Cronbach's alpha coefficient of this scale was 0.950, and the Cronbach's alpha coefficients of each dimension range from 0.70 to 0.86. In this study, Cronbach's alpha is 0.846. The scale has demonstrated good content validity and construct validity in studies involving Chinese CAD patients [30–31].

### Data collection

The Institutional Review Board of the First Affiliated Hospital of Xinjiang Medical University approved this study (K202410-20). The questionnaire survey method was employed, and the investigator introduced the purpose and significance of the study to the research participants. The questionnaire survey was conducted following the acquisition of informed consent. Data were collected, and the questionnaire was completed through one-to-one surveys for all subjects, during which the investigator asked the questions and the subjects provided their responses. The principle of voluntariness was adhered to, and informed consent was obtained from all research participants. All completed questionnaires were securely retained by the investigator.

### Data analysis

Data were entered and analysed using the SPSS 26.0 software. The measurement data were expressed as $x \pm s$ and the count data were expressed as the number of cases and percentages. Person Correlation analysis was performed to explore the relationships between the variables. The structural equation model was constructed using Mplus8.0 to test the simple mediating effect, and the mediation effect of self-efficacy between self-management and health-promoting lifestyles was assessed using the Bootstrap method at a significance level of 0.05.

## Results

### Participant characteristics

The demographics and characteristics of the participants are presented in Table 1.

### Analysis of self-management, self-efficacy, and health-promoting lifestyles

The overall self-management score was (96.66 ± 10.11), with 54 (13.50%) participants achieving a high-level score, 312 (78.00%) participants achieving a medium-level score, and 34(8.50%) participants achieving a low-level score. The total self-efficacy score was (44.49 ± 6.10), with 36 (9.00%) participants attaining a high-level score, 310 (77.50%) participants attaining a medium-level score, and 54 (13.50%) participants attaining a low-level score. The total score for health-promoting lifestyles was (162.91 ± 12.24), indicating that participants were at a favourable level overall, as demonstrated in Table 2.

### Correlation between participants' self-management behaviour, self-efficacy, and health-promoting lifestyle

The self-management behaviors, self-efficacy, and health-promoting lifestyles of participants were positively correlated with one another in pairs (all were $P < 0.01$; see Table 3), which facilitated subsequent analysis of mediation effects.

### Mediation effects

In this study, structural equation modelling was performed to examine the mediating effect of self-efficacy on the relationship between self-management and health-promoting lifestyles. A good model fit was observed for the hypothesised model. To enhance clarity, the unstandardized ($B$) and standardized ($\beta$) coefficients are distinguished and presented separately below. Fig 1 illustrates the mediation model. The path analysis indicated that self-management had a significant direct effect on health-promoting lifestyles (unstandardized $B = 0.224$, $\beta = 0.200$, $P < 0.001$). Furthermore, self-management significantly and positively predicted self-efficacy ($B = 0.307$, $\beta = 0.508$, $P < 0.001$), which in turn was a significant predictor of health-promoting lifestyles ($B = 1.164$, $\beta = 0.625$, $P < 0.001$). The bootstrap analysis (based on 5000 samples) confirmed the significance of the mediation effect. The standardized indirect effect was 0.317 (95% BCCI [0.252, 0.389]), and the direct effect was 0.200 (95% BCCI [0.121, 0.278]). The indirect effect accounted for 61.3% of the total effect. Since both the direct and indirect effects were significant, self-efficacy was identified as a partial mediator in the relationship between self-management and health-promoting lifestyles(see Table 4 and Table 5).

## Discussion

### Self-management behaviors, self-efficacy, and health-promoting behaviors

The self-management score of patients with postoperative PCI was (96.66 ± 10.11), with a score index of 71.60%, indicating a moderate level, which was higher than the findings of Zhang et al [32]. This may be due to patients losing interest in self-management over time following the initial onset of the disease, the alleviation of clinical symptoms postoperatively, and a tendency for patients to neglect follow-ups or failing to adhere to the doctor's prescriptions and medication schedules after discharge from the hospital. Additionally, some patients may not place a high value on self-management. Patients with effective self-management are motivated to regulate their behaviors to achieve desired goals or behavioural endpoints, engage in disease management actions, and modify unhealthy habits. Patients with postoperative PCI should be encouraged to adopt health management behaviors, such as adhering to medication, consuming a healthy diet, and participating in physical activity, to reduce CAD-related risk factors [33].

The self-efficacy score of patients following PCI was (44.49 ± 6.10), and the score index was 69.52%, which indicated a moderate level, and is favourable for their recovery and aligns with the studies conducted by Salgado et al [34]. This

**Table 1. Demographics and characteristics of participants.**

| Item | No. of cases | Percentage (%) |
|---|---|---|
| Sex | | |
| Male | 301 | 75.25 |
| Female | 99 | 24.75 |
| Age | | |
| <40 | 10 | 2.50 |
| 40-49 | 37 | 9.25 |
| 50-59 | 131 | 32.75 |
| 60-69 | 134 | 33.50 |
| ≥70 | 88 | 22.00 |
| Educational level | | |
| Junior high school and below | 141 | 35.25 |
| High school or junior college | 158 | 39.50 |
| Three-year college | 60 | 15.00 |
| Undergraduate and above | 41 | 10.25 |
| Marital status | | |
| Unmarried | 3 | 0.75 |
| Married | 371 | 92.75 |
| Divorced | 10 | 2.50 |
| Widowed | 16 | 4.00 |
| Monthly household income | | |
| ≤ 1000 | 18 | 4.50 |
| 1001-3000 | 28 | 7.00 |
| 3001-5000 | 105 | 26.25 |
| > 5000 | 249 | 62.25 |
| Occupational status | | |
| Employed | 126 | 31.50 |
| Retired | 183 | 45.75 |
| Unemployed | 91 | 22.75 |
| Smoking status | | |
| Yes | 138 | 34.50 |
| No | 262 | 65.50 |
| Alcohol consumption | | |
| Yes | 76 | 19.00 |
| No | 324 | 81.00 |
| Whether overweight | | |
| Yes | 276 | 69.00 |
| No | 124 | 31.00 |
| Whether this was the first diagnosis of CAD | | |
| Yes | 185 | 46.25 |
| No | 215 | 53.75 |
| Presence of comorbidities | | |
| Yes | 386 | 96.50 |
| No | 14 | 3.50 |
| Cardiac function grading | | |
| Class I | 5 | 1.25 |
| Class II | 368 | 92.00 |

*(Continued)*

**Table 1.** (Continued)

| Item | No. of cases | Percentage (%) |
|------|-------------|----------------|
| Class III | 24 | 6.00 |
| Class IV | 3 | 0.75 |

**Table 2. Description of variables.**

| Variable | Scoring range | Score | Rank |
|----------|---------------|-------|------|
| Self-management behaviors | | | |
| Total score | 26-135 | 96.66 ± 10.11 | |
| Disease knowledge management | 5-25 | 19.45 ± 3.26 | 1 |
| Emotion and cognition management | 4-20 | 17.63 ± 1.82 | 2 |
| Management of daily life | 4-20 | 16.89 ± 2.25 | 3 |
| Management of bad habits | 4-20 | 13.02 ± 3.16 | 4 |
| Symptom management | 4-20 | 12.65 ± 3.17 | 5 |
| Emergency management | 3-15 | 9.01 ± 3.19 | 6 |
| Management of treatment adherence | 2-15 | 8.02 ± 1.19 | 7 |
| Self-efficacy | | | |
| Total score | 0-64 | 44.49 ± 6.10 | |
| Control symptoms | 0-48 | 33.18 ± 4.51 | 1 |
| Maintain function | 0-16 | 11.31 ± 2.60 | 2 |
| Health-promoting lifestyle | | | |
| Total score | 52-208 | 162.91 ± 12.24 | |
| Spiritual growth | 9-36 | 35.19 ± 2.27 | 1 |
| Interpersonal relationship | 8-32 | 31.74 ± 2.07 | 2 |
| Stress management | 8-32 | 29.49 ± 3.33 | 3 |
| Nutrition | 9-36 | 27.44 ± 2.65 | 4 |
| Health responsibility | 9-36 | 23.34 ± 2.24 | 5 |
| Physical activity | 9-36 | 17.35 ± 5.70 | 6 |

**Table 3. Correlation analysis of participants' self-management behaviour, self-efficacy, and health-promoting lifestyle.**

| | Health-promoting lifestyle | Self-efficacy | Self-management |
|---|---------------------------|---------------|-----------------|
| Health-promoting lifestyle | 1 | | |
| Self-efficacy | 0.727** | 1 | |
| Self-management | 0.517** | 0.508** | 1 |

** Significant correlation at a level of 0.01 (two-tailed).

improvement may be attributed to healthcare personnel educating patients about their condition during hospitalisation or patients utilising the Internet to gain insights into their disease, thereby facilitating a preliminary understanding of their condition and treatment options. The health outcomes of treatment for all chronic diseases are significantly influenced by patients' beliefs in their ability to adhere to medication regimens and maintain a healthy lifestyle [34]. Self-efficacy, as a central element in managing negative emotions and self-managing stress, serves as a vital psychological resource that is

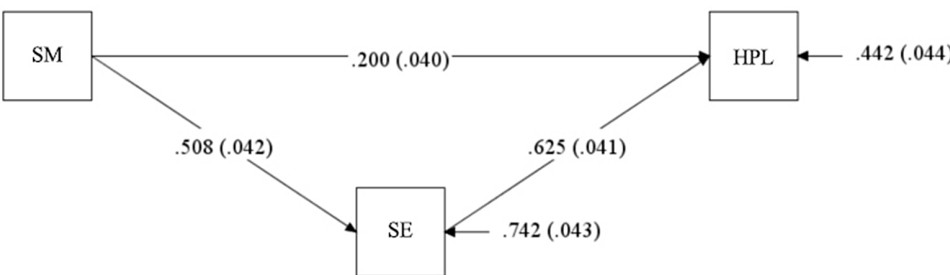

**Fig 1. Mediation model.**

**Table 4. Effect testing of the mediation model.**

| Path | B | SE | t | P-value | β | SE | t | P-value |
|------|------|------|------|------|------|------|------|------|
| SM→HPL | 0.224 | 0.045 | 4.994 | <0.01 | 0.200 | 0.040 | 5.011 | <0.01 |
| SE→HPL | 1.164 | 0.091 | 12.790 | <0.01 | 0.625 | 0.041 | 15.527 | <0.01 |
| SM→SE | 0.307 | 0.030 | 10.191 | <0.01 | 0.508 | 0.042 | 12.055 | <0.01 |

**Table 5. Bootstrap analysis for the mediation model.**

| Path | Effect (β) | SE | 95%BCCI | | 95% CI | | Mediation proportion |
|------|------|------|------|------|------|------|------|
| | | | Lower Upper | | Lower Upper | | |
| Total Effect | 0.517 | 0.042 | 0.434 | 0.595 | 0.433 | 0.595 | |
| Indirect Effect | 0.317 | 0.035 | 0.252 | 0.389 | 0.249 | 0.385 | 61.3% |
| Direct Effect | 0.200 | 0.040 | 0.121 | 0.278 | 0.120 | 0.277 | 38.7% |

*Note.* SM = Self-management; SE = Self-efficacy; HPL = Health-promoting lifestyle; *B* = Unstandardized path coefficient; *β* = Standardized path coefficient; *SE* = Standard Error; *CI* = Percentile Confidence Interval; *BCCI* = Bias-Corrected Confidence Interval.

essential for good prognosis and quality of life in patients [35]. It is crucial for patients to develop a sense of self-efficacy to assist them in coping with adversity and challenges [36].

The health-promoting lifestyle score of patients following PCI was (162.91 ± 12.24), which is favourable and aligns with the studies conducted by Yun et al and Hajizadeh-Sharafabad et al [37–38]. Regarding self-fulfilment, interpersonal relationships, and stress management, it is possible that a significant proportion of the study population was nearing retirement or had already retired. Consequently, they possessed rich life experiences and a broad range of social connections, enabling them to typically establish strong interpersonal relationships with those around them. Furthermore, their work was less demanding, resulting in minimal work-related pressure. Additionally, it is plausible that during this stage, they experienced low levels of pressure from both life and work, which facilitated emotional, information, material, or practical support. In relation to neglecting nutrition, health responsibilities, and physical activity, it is possible that the patient's fear of incurring excessive expenses for hospital visits, combined with the symptomatic relief experienced postoperatively, has resulted in poor compliance and negligence regarding medication and follow-ups.

Regarding their diet, patients tended to favour carbohydrate-rich foods while disregarding the importance of balanced nutrition. Younger patients often feel too busy with work to engage in exercise or believe they are in good physical shape, thereby failing to prioritise physical activity. As patients age, a decline in bodily functions or the onset of comorbidities may contribute to a reduction in exercise levels.

## Mediating role of self-efficacy between self-management and health-promoting behaviors

In this study, structural equation modelling was performed to construct the pathways of action involving self-management, self-efficacy, and health-promoting behaviors. The results indicated that self-management positively predicted a health-promoting behaviors; specifically, higher levels of self-management were associated with improved health-promoting behaviors outcomes. Furthermore, self-efficacy was found to positively predict a health-promoting behaviors, with greater levels of self-efficacy linked to improved psychological states and enhanced health-promoting behaviors. Within the context of this single-center study in China, Self-efficacy serves as a mediating variable, indirectly influencing a health-promoting behaviors. The indirect effect accounted for 61.3% of the total effect. This substantial proportion indicates that self-efficacy is not merely a minor contributor but a central psychological pathway through which self-management influences lifestyle. Clinically, this underscores that interventions solely focusing on knowledge provision may be insufficient; actively targeting and enhancing self-efficacy could potentially amplify the effectiveness of cardiac rehabilitation programs by more than half. This transformation can be understood through Bandura's theory [39]: self-efficacy likely operates by enhancing patients' perceived capability to overcome barriers, fostering positive outcome expectancies for health behaviors, and increasing resilience in the face of setbacks, thereby bridging the intention-action gap. Consequently, to improve patients' self-management capabilities, healthcare professionals should also focus on improving patients' self-efficacy and provide guidance to further support the health-promoting lifestyles of patients post-PCI.

This suggests that healthcare providers can adopt effective measures to improve patients' self-management and self-efficacy levels, thereby enhancing health-promoting behaviors and facilitating the transition from merely understanding the disease to effectively controlling it. Individuals with high self-efficacy are generally more capable of overcoming challenges and adhering to self-discipline than those with low self-efficacy. Furthermore, those with high self-efficacy tend to cope with discomfort using a positive mindset and experience fewer negative emotions, as well as possess greater confidence in overcoming disease, resulting in an enhanced sense of conviction, well-being, and fulfilment in life [29]. Healthcare professionals can enhance patients' confidence in performing health behaviors by strengthening their ability to understand and control the disease and introduce patients to the precautions following PCI in a clear and comprehensible manner, develop personalised diet plans tailored to individual needs, and educate patients on how to read ingredient lists on food labels to manage their calorie and nutrient intake effectively. Furthermore, healthcare professionals should design appropriate exercise programmes based on the patient's physical condition, including cardiac function, and instruct patients on how to recognise potential adverse symptoms that may occur postoperatively, such as chest pain and dyspnoea, along with the necessary measures to take in response to these symptoms, such as resting immediately and contacting healthcare personnel.

Additionally, healthcare professionals should conduct regular follow-ups with patients and acknowledge their progress in self-management to enhance their confidence. They should also assist patients in setting short-term and long-term health goals and refine these goals to facilitate step-by-step achievement.

## Limitations

Several limitations of this study should be considered. First, participants were recruited from a Grade 3 hospital in Xinjiang, China. Therefore, the participant information may not be generalisable to those from hospitals in other cities or at the grassroot-level. Future studies should aim to recruit participants from hospitals across the country. Second, the internal consistency of the Cardiac Self-Efficacy Scale in this sample ($\alpha = 0.640$) was lower than that of the original scale, which may affect the robustness of the mediation analysis involving this construct. This lower reliability may have attenuated the observed relationships involving self-efficacy, suggesting that the true mediating effect could be stronger than estimated. Future studies using scales with higher internal consistency in this population are needed to confirm our findings. Third, only one variable, namely self-efficacy, was included as a mediating variable in this study. It is recommended that additional variables be incorporated in follow-up studies to enhance the credibility of the findings. Fourth, the cross-sectional

design limits causal inference. While the proposed mediation model is theoretically grounded, longitudinal or interventional studies are needed to confirm the temporal and causal relationships between self-management, self-efficacy, and health-promoting behaviors.

## Conclusion

Both the self-management and self-efficacy levels of patients post-PCI can positively predict health-promoting behaviors, with self-efficacy playing a partial mediating role. This suggests that clinical interventions should not only provide knowledge and skills training but also actively incorporate strategies to build patients' confidence in their ability to execute and sustain these behaviors. Practical strategies may include guided mastery experiences, positive feedback from healthcare providers, and peer support groups. Furthermore, the sociodemographic characteristics of the sample, such as age, educational level, and employment status, highlight the need for personalized intervention approaches. This suggests that healthcare professionals should prioritise the self-management and self-efficacy levels of patients. Furthermore, targeted interventions may be implemented to promote healthy behaviors, such as a balanced diet, regular physical exercise, disease monitoring, and rational medication use. These interventions aim to enhance the self-management and self-efficacy of patients, thereby improving their health-promoting behaviors and long-term cardiovascular outcomes. In this population. Future interventions should integrate confidence-building strategies and be tailored to patients' sociodemographic backgrounds to more effectively translate self-management knowledge into sustained action.

## Author contributions

**Data curation:** Ping Yan, Fang Hou.

**Formal analysis:** Xin Gu, Lina Ma.

**Funding acquisition:** li Zhang.

**Investigation:** Zhijie Cao, Hu Peng.

**Project administration:** li Zhang.

**Writing – original draft:** Zhijie Cao.

**Writing – review & editing:** Xin Gu, li Zhang.

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
