## [Decision Letter · Decision Letter 0]

17 Nov 2025

Dear Dr. Zhang,

Reviewers reported major issues concerning statistics and methods. Please provided a point to point reply to the reviewers' comments.

We look forward to receiving your revised manuscript.

Kind regards,

Chiara Lazzeri

Academic Editor

PLOS ONE

Journal Requirements:

2. In the ethics statement in the Methods, you have specified that verbal consent was obtained. Please provide additional details regarding how this consent was documented and witnessed, and state whether this was approved by the IRB.

“This work were supported by the Special Nursing Project of the State Key Laboratory of Pathogenesis, Prevention and Treatment of High Incidence Diseases in Central Asia, a provincial-ministry joint project (SKL-HIDCA-2023-HL4),Xinjiang Key Laboratory of Medical Animal Mode Research and Excellent Talents Innovative Teams of the First Affiliated Hospital of Xinjiang Medical University (cxtd202414).”

“This work were supported by the Special Nursing Project of the State Key Laboratory of Pathogenesis, Prevention and Treatment of High Incidence Diseases in Central Asia, a provincial-ministry joint project (SKL-HIDCA-2023-HL4),Xinjiang Key Laboratory of Medical Animal Mode Research and Excellent Talents Innovative Teams of the First Affiliated Hospital of Xinjiang Medical University (cxtd202414).”

5. In the online submission form, you indicated that [The data underlying the results presented in the study are available from the corresponding author.].

Reviewers' comments:

Reviewer's Responses to Questions

**Comments to the Author**

1. Is the manuscript technically sound, and do the data support the conclusions?

Reviewer #1: Partly

Reviewer #2: Partly

2. Has the statistical analysis been performed appropriately and rigorously?

Reviewer #1: Yes

Reviewer #2: Yes

3. Have the authors made all data underlying the findings in their manuscript fully available?

Reviewer #1: No

Reviewer #2: Yes

4. Is the manuscript presented in an intelligible fashion and written in standard English?

Reviewer #1: Yes

Reviewer #2: Yes

Reviewer #1: This article investigates the mediating role of self-efficacy in the relationship between self-management and health-promoting behaviors in post-PCI patients. The topic is timely and relevant, particularly in the context of secondary prevention of coronary artery disease. The manuscript is generally well-structured and offers clear clinical implications.

However, several sections of the manuscript would benefit from clarification, elaboration, or correction. Below are specific comments organized by manuscript section:

Measurement reliability

Although the original Cronbach’s alpha for the Cardiac Self-Efficacy Scale was acceptable (α = 0.820), the coefficient obtained in the current study was notably lower (α = 0.640), suggesting limited internal consistency in this sample. This should be acknowledged as a limitation and discussed in relation to the robustness of the mediation analysis.

Data inconsistency (Table 2):

The mean score for health-promoting lifestyle reported in the text (164.57 ± 11.36) does not match the value presented in Table 2 (162.91 ± 12.24). Please verify and ensure consistency between text and tables.

Mediation model description (Table 4)

The section describing the mediation model includes several inconsistencies. The text cites a direct path coefficient of γ = 0.200, while Table 4 reports the corresponding unstandardized effect as 1.164. This likely reflects confusion between standardized path coefficients (γ) and unstandardized estimates. Additionally, the value γ = 0.508 is cited for both the self-management → self-efficacy path and the self-efficacy → health-promoting behaviors path, which seems implausible and may be a copy-paste or labeling error. Moreover, no standardized path coefficients (γ) are actually presented in any table or figure, making it difficult for readers to interpret the reported values. To improve clarity, I recommend that the authors:

• Clearly distinguish between standardized coefficients (γ) and unstandardized effects throughout the manuscript;

• Include a separate table reporting all standardized path coefficients with their standard errors and significance levels;

• Avoid referencing numerical values in the narrative that are not verifiable from the presented tables or figures.

Discussion

I recommend formulating generalizations about the role of self-efficacy with more caution, considering that the study was conducted at a single center in China and involved a specific sample of post-PCI patients.

Conclusion

The “Conclusion” section provides a coherent summary of the study’s findings and their clinical implications. However, it would be beneficial to expand this section by emphasizing the mediating role of self-efficacy in the context of practical interventions, as well as by considering the impact of sociodemographic characteristics analyzed in the study.

Reviewer #2: Reviewer Comment: The mediating role of self-efficacy in the relationship between self-management and health-promoting behaviors in post-PCI patients

Typos/Grammatical Errors:

• Page 1, Line 17: Remove “a”: “health promoting behaviors

• Page 4, Line 111-113: Sentences are fragmented due to line breaks (e.g., “behaviours.The Cronbach’s alpha...”)

• Page 11, Line 224: “wh ich is favourable”

• Page 14, Line 258: “patien ts'”

• Page 16, Line 292: “patients. Furthermore, targeted interventions may be implemented…”- Combine for better flow of the sentence

Narrative:

• Page 2-3, Line 36-84: The introduction effectively presents the rationale but could be tightened for clarity. Several ideas overlap (e.g., repetitive mention of CAD burden and PCI limitations).

• Page 11-13, Line 199-257: The discussion section largely restates results rather than interpreting them deeply. It would benefit from connecting findings to specific mechanisms (e.g., how self-efficacy transforms motivation into lifestyle adherence).

• Page 15, Line 281-289: Well-structured limitation portion but needs more analytical reflection.

• Page 13-15: Effect sizes (γ-values) and mediation proportion should be discussed more critically. Eg disucss why 23.5% mediation is clinically meaningful

Methods/Results:

• Page 4-6: Sampling and scale sections are clear but lack justification for validity

• Page 9-10: Data representations seems to be repetitive between text and tables

what does this mean?). If published, this will include your full peer review and any attached files.

**Do you want your identity to be public for this peer review?** For information about this choice, including consent withdrawal, please see our Privacy Policy

Reviewer #1: No

Reviewer #2: No

---

## [Author Response · Author response to Decision Letter 1]

7 Jan 2026

Dear Editor

Thank you for your decision letter concerning our manuscript entitled "The mediating role of self-efficacy in the relationship between self-management and health-promoting behaviors in post-PCI patients" and for carefully reviewing our manuscript. We also appreciate all the critical comments from you and the reviewers. We have carefully considered the comments and revised the manuscript again accordingly. With these improvements, we hope that the current version can meet the Journal’s standards for publication. The following is a point-by-point response to all those comments and a list of changes we have made to the manuscript.

Point-by-point responses to the comments of the Editor and reviewers and a list of changes are:

Editorial

Dear Editor,

During the revision of this manuscript, in accordance with the requirements for the formal closure of the funding project supporting this research (Grant No.: SKL-HIDCA-2023-HL4), we have updated the Authorship Affiliation Statement and the Funding Statement accordingly.

The specific adjustments are as follows:

Added the Department of Nursing, The First Affiliated Hospital of Xinjiang Medical University, and the State Key Laboratory of Pathogenesis, Prevention and Treatment of High Incidence Diseases in Central Asia as collaborative research institutions.

Adjusted the order of the listed author affiliations and funding bodies to accurately align with the administrative and reporting structure required by the funding project.

We wish to emphasize that these adjustments solely involve the administrative section of the manuscript. They do not entail any modifications to the core academic content, including the study’s methodology, results, discussion, or conclusions, and the scientific content of the manuscript remains unchanged.

We sincerely apologize for any inconvenience caused by this administrative adjustment and thank you for your understanding.

Reviewer 1

Q1: Although the original Cronbach’s alpha for the Cardiac Self-Efficacy Scale was acceptable (α = 0.820), the coefficient obtained in the current study was notably lower (α = 0.640), suggesting limited internal consistency in this sample. This should be acknowledged as a limitation and discussed in relation to the robustness of the mediation analysis.

A1: We thank the reviewer for this important methodological point. We have now explicitly acknowledged and discussed this limitation in the “Limitations” section of the manuscript. Specifically, we noted that the lower internal consistency may attenuate the observed relationships involving self-efficacy and that future studies using scales with higher internal consistency are needed to confirm our findings.(Page 13, lines 303-308 in the revised version).

Q2: The mean score for health-promoting lifestyle reported in the text (164.57 ± 11.36) does not match the value presented in Table 2 (162.91 ± 12.24). Please verify and ensure consistency between text and tables.

A2: We apologize for this inconsistency. The correct value is (162.91±12.24), which is now consistently reported in both the text and Table 2. The text has been updated accordingly.

Q3: The section describing the mediation model includes several inconsistencies. The text cites a direct path coefficient of γ = 0.200, while Table 4 reports the corresponding unstandardized effect as 1.164. This likely reflects confusion between standardized path coefficients (γ) and unstandardized estimates. Additionally, the value γ = 0.508 is cited for both the self-management → self-efficacy path and the self-efficacy → health-promoting behaviors path, which seems implausible and may be a copy-paste or labeling error. Moreover, no standardized path coefficients (γ) are actually presented in any table or figure, making it difficult for readers to interpret the reported values. To improve clarity, I recommend that the authors:

• Clearly distinguish between standardized coefficients (γ) and unstandardized effects throughout the manuscript;

• Include a separate table reporting all standardized path coefficients with their standard errors and significance levels;

• Avoid referencing numerical values in the narrative that are not verifiable from the presented tables or figures.

A3: We thank the reviewer for pointing out these inconsistencies and for the helpful suggestions. We have thoroughly revised the mediation model presentation to enhance clarity: We have clearly distinguished between unstandardized (B) and standardized (β) coefficients in the text and tables; Table 4 now presents both unstandardized (B) and standardized (β) coefficients for all paths, along with their standard errors and significance levels; We have removed any ambiguous references to “γ” and consistently used β for standardized coefficients in the narrative; The text now explicitly refers to Table 4 and Table 5 for the exact coefficients, avoiding any unattributed numerical claims.(Page 9, lines 195-207 and page 9, lines 209-212 in the revised version).

Q4:I recommend formulating generalizations about the role of self-efficacy with more caution, considering that the study was conducted at a single center in China and involved a specific sample of post-PCI patients.

A4: We agree with the reviewer and have moderated our generalizations about self-efficacy. In the Discussion and Conclusion, we now emphasize that our findings are based on a single-center sample in China and that future multi-center or cross-cultural studies are needed to confirm the generalizability of the results.(Page 12, line 264 in the revised version).

Q5:The “Conclusion” section provides a coherent summary of the study’s findings and their clinical implications. However, it would be beneficial to expand this section by emphasizing the mediating role of self-efficacy in the context of practical interventions, as well as by considering the impact of sociodemographic characteristics analyzed in the study.

A5: We thank the reviewer for this constructive suggestion. We have expanded the Conclusion to more explicitly highlight the mediating role of self-efficacy in practical interventions and to incorporate the sociodemographic characteristics. (Page 14, lines 318-324 in the revised version).

Reviewer 2

Q1: Typos/Grammatical Errors:

• Page 1, Line 17: Remove “a”: “health promoting behaviors

• Page 4, Line 111-113: Sentences are fragmented due to line breaks (e.g., “behaviours.The Cronbach’s alpha...”)

• Page 11, Line 224: “wh ich is favourable”

• Page 14, Line 258: “patien ts'”

• Page 16, Line 292: “patients. Furthermore, targeted interventions may be implemented…”- Combine for better flow of the sentence

A1: We thank the reviewer for catching these language errors. We have carefully proofread the manuscript and corrected all identified typos and grammatical issues.

Q2: Narrative:

• Page 2-3, Line 36-84: The introduction effectively presents the rationale but could be tightened for clarity. Several ideas overlap (e.g., repetitive mention of CAD burden and PCI limitations).

• Page 11-13, Line 199-257: The discussion section largely restates results rather than interpreting them deeply. It would benefit from connecting findings to specific mechanisms (e.g., how self-efficacy transforms motivation into lifestyle adherence).

• Page 15, Line 281-289: Well-structured limitation portion but needs more analytical reflection.

• Page 13-15: Effect sizes (γ-values) and mediation proportion should be discussed more critically. Eg disucss why 23.5% mediation is clinically meaningful

A2: We appreciate the reviewer’s suggestions to enhance the narrative depth and critical reflection. We have made the following revisions:

Introduction: We have tightened the introduction by merging overlapping statements and removing repetitive mentions of CAD burden and PCI limitations, improving logical flow.(Page 2, lines 40-41 and 47-49 in the revised version).

Discussion: We have deepened the interpretation by explicitly linking self-efficacy to behavior-change mechanisms and explaining how self-efficacy bridges the intention-action gap in health behavior adoption.(Page 12, lines 265-274 in the revised version).

Limitations: We have added more analytical reflection, particularly on the implications of the lower Cronbach’s alpha for the self-efficacy scale and the cross-sectional design.(Page 13, lines 304-309 and 312-314 in the revised version).

Effect sizes: We now discuss the clinical meaningfulness of the mediation proportion (61.3% of the total effect), emphasizing that self-efficacy is a central psychological pathway in translating self-management into behavior change.(Page 12, lines 265-274 in the revised version).

Q3: Methods/Results:

• Page 4-6: Sampling and scale sections are clear but lack justification for validity

• Page 9-10: Data representations seems to be repetitive between text and tables

A3:We sincerely appreciate the reviewer's valuable suggestions, we have addressed these two points as follows:

Validity justification: We have added a brief justification for the use of each scale, noting that they have been previously validated in Chinese CAD populations and citing relevant studies.(Page 5, lines 125-127 and 137-139, page 6, lines 149-151 in the revised version).

Redundancy between text and tables: We have streamlined the text in the Results section to avoid repeating information that is clearly presented in the tables. For example, we now refer readers to Table 1 for detailed participant characteristics rather than repeating all statistics in the text.(Page 6, line 171 in the revised version).

Once again, thank you for your valuable suggestion. We look forward to hearing from you.

Sincerely,

Li Zhang

The First Affiliated Hospital of Xinjiang Medical University,

No. 393, Liyue Mountain Avenue, Urumqi 830000, China.

Email: 15684885009@163.com

---

## [Editor Report · Decision Letter 1]

13 Jan 2026

The mediating role of self-efficacy in the relationship between self-management and health-promoting behaviors in post-PCI patients

PONE-D-25-30248R1

Dear Dr. Zhang,

We’re pleased to inform you that your manuscript has been judged scientifically suitable for publication and will be formally accepted for publication once it meets all outstanding technical requirements.

Kind regards,

Chiara Lazzeri

Academic Editor

PLOS One
---

## [Editor Report · Acceptance letter]

PONE-D-25-30248R1

PLOS One

Dear Dr. Zhang,

I'm pleased to inform you that your manuscript has been deemed suitable for publication in PLOS One. Congratulations! Your manuscript is now being handed over to our production team.

Kind regards,

on behalf of

Dr. Chiara Lazzeri

Academic Editor

PLOS One